REPORT

# Definition of phosphatidylinositol 4,5-bisphosphate distribution by freeze-fracture replica labeling

Takuma Tsuji[1], Junya Hasegawa[2], Takehiko Sasaki[2], and Toyoshi Fujimoto[1]

**Phosphatidylinositol 4,5-bisphosphate [PtdIns(4,5)P$_2$] is a phospholipid essential for plasma membrane functions, but its two-dimensional distribution is not clear. Here, we compared the result of sodium dodecyl sulfate–treated freeze-fracture replica labeling (SDS-FRL) of quick-frozen cells with the actual PtdIns(4,5)P$_2$ content and the results obtained by fluorescence biosensor and by labeling of chemically-fixed membranes. In yeast, enrichment of PtdIns(4,5)P$_2$ in the membrane compartment of Can1 (MCC)/eisosome, especially in the curved MCC/eisosome, was evident by SDS-FRL, but not by fluorescence biosensor, GFP-PLC1δ-PH. PtdIns(4,5)P$_2$ remaining after acute ATP depletion and in the stationary phase, 30.0% and 56.6% of the control level, respectively, was not detectable by fluorescence biosensor, whereas the label intensity by SDS-FRL reflected the PtdIns(4,5)P$_2$ amount. In PC12 cells, PtdIns(4,5)P$_2$ was observed in a punctate pattern in the formaldehyde-fixed plasma membrane, whereas it was distributed randomly by SDS-FRL and showed clustering after formaldehyde fixation. The results indicate that the distribution of PtdIns(4,5)P$_2$ can be defined most reliably by SDS-FRL of quick-frozen cells.**

## Introduction

Phosphatidylinositol 4,5-bisphosphate [PtdIns(4,5)P$_2$] is a major phosphoinositide in the plasma membrane and plays essential roles in signaling, cytoskeletal anchorage, regulation of ion channels and transporters, endocytosis, and exocytosis, among others (Balla, 2013; Dickson and Hille, 2019). How PtdIns(4,5)P$_2$ can exert many different functions without mutual interference is not clear, but the regional compartmentalization of PtdIns(4,5)P$_2$ through interactions with proteins and lipids has been hypothesized (Hilgemann, 2007; Hope and Pike, 1996; van den Bogaart et al., 2011; Wen et al., 2018). To test this hypothesis, it is critical to know how PtdIns(4,5)P$_2$ distributes two dimensionally in the membrane, but this has not been easy due to technical difficulties.

In the plasma membrane of budding yeast, a membrane domain called the membrane compartment of Can1 (MCC) is stabilized by the eisosome, a self-assembly of PtdIns(4,5)P$_2$-binding proteins, Pil1 and Lsp1 (Karotki et al., 2011). MCC/eisosome also recruits PtdIns(4,5)P$_2$-binding Slm1 and Slm2 (Olivera-Couto et al., 2011), and PtdIns(4,5)P$_2$-phosphatase, Inp51 (Fröhlich et al., 2014), suggesting that the MCC/eisosome domain contains abundant PtdIns(4,5)P$_2$. Fluorescence biosensors for PtdIns(4,5)P$_2$, e.g., GFP-tagged pleckstrin homology domain of phospholipase Cδ1 (GFP-PH$^{PLCδ1}$), however, do not show any denser distribution in MCC/eisosome than in other plasmalemmal areas (Spira et al., 2012) (Fig. S1 A). This result may simply indicate that PtdIns(4,5)P$_2$ is not enriched in MCC/eisosome, but it is more likely that biosensors do not bind to PtdIns(4,5)P$_2$ in MCC/eisosome efficiently (Fröhlich et al., 2014).

In the plasma membrane of PC12 cells, a neuronal cell model, constitutive co-clustering of PtdIns(4,5)P$_2$ and syntaxin-1A was suggested by fluorescence labeling of formaldehyde-fixed membrane sheets (Fig. S1 B) (van den Bogaart et al., 2011). The putative PtdIns(4,5)P$_2$–syntaxin-1A co-cluster has been thought to function as docking sites for synaptic vesicles in the presynaptic membrane, but it is not clear whether PtdIns(4,5)P$_2$ is actually clustered in living cells because the distribution might be changed by the labeling procedure, which includes cooling, unroofing (i.e., mechanical disruption of cells), chemical fixation, and ligand binding.

In defining membrane lipid distribution, sodium dodecyl sulfate–treated freeze-fracture replica labeling (SDS-FRL) has advantages over the fluorescence biosensor method and the membrane sheet method because it does not require the expression of biosensors in live cells or treatments that may affect lipid distribution (Tsuji et al., 2019b). In SDS-FRL, phospholipids physically fixed by the freeze-fracture replica, a solid made of platinum and carbon, are likely retained at the locations they occupied at the moment of quick-freezing, whereas cytosolic

---

[1]Research Institute for Diseases of Old Age, Juntendo University Graduate School of Medicine, Tokyo, Japan; [2]Medical Research Institute, Tokyo Medical and Dental University, Tokyo, Japan.

Correspondence to Toyoshi Fujimoto: t.fujimoto.xl@juntendo.ac.jp.

and peripheral membrane proteins adhering to the surface of freeze-fracture replicas are removed by SDS treatment (Fujimoto et al., 1996; Fujita and Fujimoto, 2007). Further treatment of the freeze-fracture replica with proteinase K can excise extramembrane portions of integral membrane proteins, improving the accessibility of probes to target lipids (Fig. S1 C). Although the advantages of SDS-FRL are theoretically evident, it is still uncertain whether the result accurately reflects the actual amount of target lipids and how they compare with results obtained by other methods. In this study, we addressed these points using yeast and mammalian plasma membranes.

First, using budding yeast, PtdIns(4,5)P$_2$ was quantified by phosphoinositide regioisomer measurement by chiral column chromatography and mass spectrometry (PRMC-MS), which can distinguish eight classes of phosphoinositides (Morioka et al., 2022), and the result was compared with the labeling obtained by SDS-FRL and the fluorescence biosensor method. The PRMC-MS result agreed well with that of SDS-FRL, but not with that of the biosensor method, and the defect of the latter was found to be derived from the poor recognition of PtdIns(4,5)P$_2$, not only in MCC/eisosome but also in the non-MCC area. Second, in PC12 cells, clustering of the PtdIns(4,5)P$_2$ labels, which was observed by the membrane sheet labeling method was not reproduced in SDS-FRL of quick-frozen cells. By using cells that were fixed before quick-freezing, formaldehyde fixation was found to induce the clustering of PtdIns(4,5)P$_2$ labels, at least partially due to the obstruction of the access of labeling probes to PtdIns(4,5)P$_2$ by crosslinked proteins. These results corroborate that the two-dimensional distribution of PtdIns(4,5)P$_2$ can be defined most reliably by SDS-FRL of quick-frozen cells.

## Results and discussion

### PtdIns(4,5)P$_2$ distribution in yeast plasma membrane

MCC/eisosome in the plasma membrane of *Saccharomyces cerevisiae* is a longitudinal furrow-like domain of ~300 nm in length and 50 nm in depth (Douglas and Konopka, 2014; Ziółkowska et al., 2012). Electron tomography revealed that most MCC/eisosome furrows are highly curved, whereas others are shallow, and they were thus named curved and shallow MCC/eisosomes, respectively (Bharat et al., 2018). Furrows of different depths were also observed in freeze-fracture replicas (Fig. 1 A). By SDS-FRL, Sur7-GFP was labeled in those furrows and Pma1-GFP was excluded, verifying that they are MCC/eisosomes (Malínská et al., 2003; Malinska et al., 2004; Young et al., 2002) (Fig. 1 B).

GFP-PH$^{PLC\delta1}$ has been employed as a fluorescence biosensor for PtdIns(4,5)P$_2$ (Stauffer et al., 1998; Várnai and Balla, 1998). In log-phase yeast, linear fluorescence of GFP-PH$^{PLC\delta1}$ was observed along the cell periphery, indicating the presence of PtdIns(4,5)P$_2$ in the plasma membrane, but it did not show a local concentration indicative of enrichment in MCC/eisosome (Fig. 1 C). This is not because the size of MCC/eisosome is too small for diffraction-limited fluorescence microscopy, since Pil1-GFP, an eisosome component, was observed as distinct puncta (Fig. S2 A). Even by microscopy with a higher

space resolution, GFP-PH$^{PLC\delta1}$ was not observed as puncta (Spira et al., 2012).

Using SDS-FRL, PtdIns(4,5)P$_2$, labeled with glutathione S-transferase (GST)-tagged PH$^{PLC\delta1}$ (Fujita et al., 2009), was found to be distributed most densely in the curved MCC/eisosome, followed by the shallow MCC/eisosome and non-MCC areas (Fig. 1, D and E; and Fig. S2 B). The accumulation of PtdIns(4,5)P$_2$ in MCC/eisosomes was also evident when the label distribution was mapped according to the distance from the center line of MCC/eisosomes (Fig. 1 F). The specificity of the labeling was verified by the virtual absence of labels when GST-PH$^{PLC\delta1(K30N,K32N)}$, a mutant lacking binding affinity for PtdIns(4,5)P$_2$ (Stauffer et al., 1998), was used instead of GFP-PH$^{PLC\delta1}$ (Fig. 1 G). For quantification of the PtdIns(4,5)P$_2$ label density, (1) SDS-treated freeze-fracture replicas were digested with proteinase K to cleave extramembrane portions of transmembrane proteins, ensuring probe access to PtdIns(4,5)P$_2$ (Fig. S1 C), and (2) GFP-PH$^{PLC\delta1}$ was used at a low concentration (12.5 ng/ml) to avoid saturation of labeling caused by steric hindrance between probes (Fig. S2 C) (Fujita et al., 2009). The results obtained showed a small cell-to-cell variation in the PtdIns(4,5)P$_2$ label density, verifying the robustness of data obtained by SDS-FRL (Fig. 1 H).

The above result indicated that PtdIns(4,5)P$_2$ is highly enriched in MCC/eisosomes. However, there remained a possibility that PtdIns(4,5)P$_2$ bound to endogenous proteins, such as Pil1 and Lsp1 (Karotki et al., 2011), might be labeled more efficiently than free PtdIns(4,5)P$_2$ in SDS-FRL, causing the dense labeling in MCC/eisosomes. We addressed this possibility by labeling PtdIns(4,5)P$_2$ in liposomes, which were incubated with or without recombinant GFP-PH$^{PLC\delta1}$ before quick-freezing (Fig. S2 D). The result showed that PtdIns(4,5)P$_2$ in the two groups of liposomes was labeled in comparable intensities by SDS-FRL (Fig. S2 E), confirming that the high label density in the MCC/eisosome reflects the actual enrichment of PtdIns(4,5)P$_2$ in that domain.

We next analyzed whether the PtdIns(4,5)P$_2$ amount quantified by PRMC-MS is correlated with the PtdIns(4,5)P$_2$ label intensity obtained by the fluorescence biosensor method and SDS-FRL. For this purpose, yeast was subjected to two conditions that decrease PtdIns(4,5)P$_2$: one is acute ATP depletion by treatment with 2% 2-deoxyglucose and 3 mM sodium azide for 5 min, and the other is slow ATP depletion by nutrient exhaustion and induction of the stationary phase. By ATP depletion, PtdIns(4,5)P$_2$ decreases by the arrest of synthesis and the hydrolysis of existing PtdIns(4,5)P$_2$ (Riggi et al., 2018).

After acute ATP depletion, the cellular PtdIns(4,5)P$_2$ content measured by PRMC-MS was decreased to 30.0% of the control level (Fig. 2 A). Here, the PtdIns(4,5)P$_2$ content obtained by PRMC-MS is derived not only from the plasma membrane but also from intracellular organelles (Tan et al., 2015), but intracellular PtdIns(4,5)P$_2$ is thought to be much less than that in the plasmalemmal pool (Vicinanza et al., 2011; Watt et al., 2002) and is likely to decrease upon ATP depletion as well. Thus, we reasoned that the relative decrease of PtdIns(4,5)P$_2$ in the plasma membrane is similar to that in the total cell.

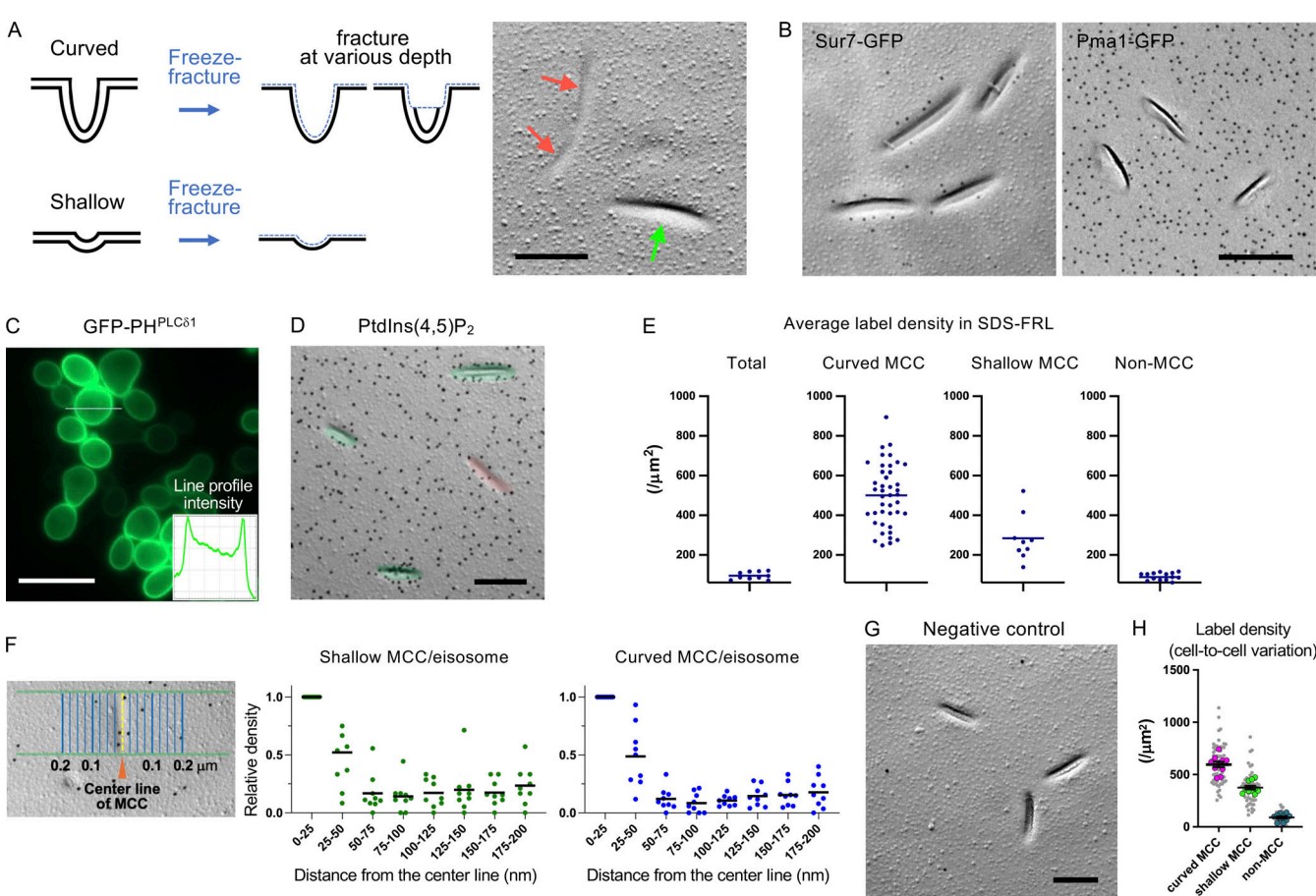

Figure 1. **PtdIns(4,5)P₂ in log phase yeast. (A)** Freeze-fracture images of curved MCC (green) and shallow MCC (red). Bar, 0.1 μm. **(B)** Distribution of Sur7-GFP and Pma1-GFP shown by SDS-FRL. Bar, 0.2 μm. **(C)** PtdIns(4,5)P₂ distribution indicated by fluorescence biosensor GFP-PH$^{PLCδ1}$. Bar, 10 μm. **(D)** PtdIns(4,5)P₂ distribution shown by SDS-FRL. Curved MCC (green) and shallow MCC (red) are colored. Bar, 0.2 μm. **(E)** The PtdIns(4,5)P₂ label density by SDS-FRL. The results in the total cell membrane (*n* = 10), curved MCC (*n* = 66), shallow MCC (*n* = 9), and non-MCC areas (*n* = 15) are shown. **(F)** The PtdIns(4,5)P₂ label density was measured at intervals of 25 nm from the center line of MCC/eisosomes. Left: schematic of the method. Right: results for shallow and curved MCC/eisosomes. The number of MCC/eisosomes measured: 69 (shallow), 71 (curved). **(G)** Yeast replica labeled with a mutant probe, GST-PH$^{PLCδ1(K30N,K32N)}$. Bar, 0.2 μm. **(H)** The density of PtdIns(4,5)P₂ labels in different cells. The colored circles indicate the average density of the labels in each cell. Number of cells counted: 10.

In this condition, the fluorescence intensity of the biosensor GFP-PH$^{PLCδ1}$ at the cell surface was indistinguishable from that in the cytoplasm (Fig. 2 B). The cell surface labeling quantified by the relative ratio of the fluorescence intensity at the cell surface to that in the cytoplasm (relative F$_{PM}$) (Nishimura et al., 2019) decreased to ∼1, the theoretical lower limit by this method of quantification (Fig. 2 B). By SDS-FRL, the total PtdIns(4,5)P₂ label decreased to 40.2% of the control (Fig. 2, C and D). The average label density in the non-MCC area decreased drastically, whereas that in the curved and shallow MCCs exhibited a milder decrease and an insignificant decrease, respectively. Because the total number of the PtdIns(4,5)P₂ label in the non-MCC area is larger than that in MCC/eisosome, the decrease after acute ATP depletion is attributable to that in the non-MCC area for the most part (Fig. 2 D).

Next, yeast in the stationary phase was analyzed for PtdIns(4,5)P₂ content by PRMC-MS and the label intensity by the fluorescence biosensor method and by SDS-FRL. The PtdIns(4,5)P₂ content in this condition was 56.6% of the control

yeast (Fig. 3 A), nearly twofold more than that after acute ATP depletion. Nevertheless, the GFP-PH$^{PLCδ1}$ fluorescence at the cell surface could not be distinguished from that in the cytoplasm, so that relative F$_{PM}$ was at the basal level (Fig. 3 B).

In the stationary phase, the MCC/eisosome domain expands (Gournas et al., 2018). This was also observed in freeze-fracture micrographs, which revealed marked elongation of MCC/eisosome furrows and a significant increase of the total MCC area (10.0% in the stationary phase versus 4.5% in the log phase) (Fig. 3 C). In this condition, the total PtdIns(4,5)P₂ label by SDS-FRL was 58.0% of the control level (Fig. 3 D), in good agreement with the actual PtdIns(4,5)P₂ amount. Here, although the average labeling density in the curved MCC/eisosome decreased significantly in the stationary phase, the total number of PtdIns(4,5)P₂ label in MCC/eisosome increased because of its area expansion, whereas that in the non-MCC area decreased (Fig. 3, C and D).

These results showed that the labeling of PtdIns(4,5)P₂ by SDS-FRL occurs roughly in accordance with the actual content,

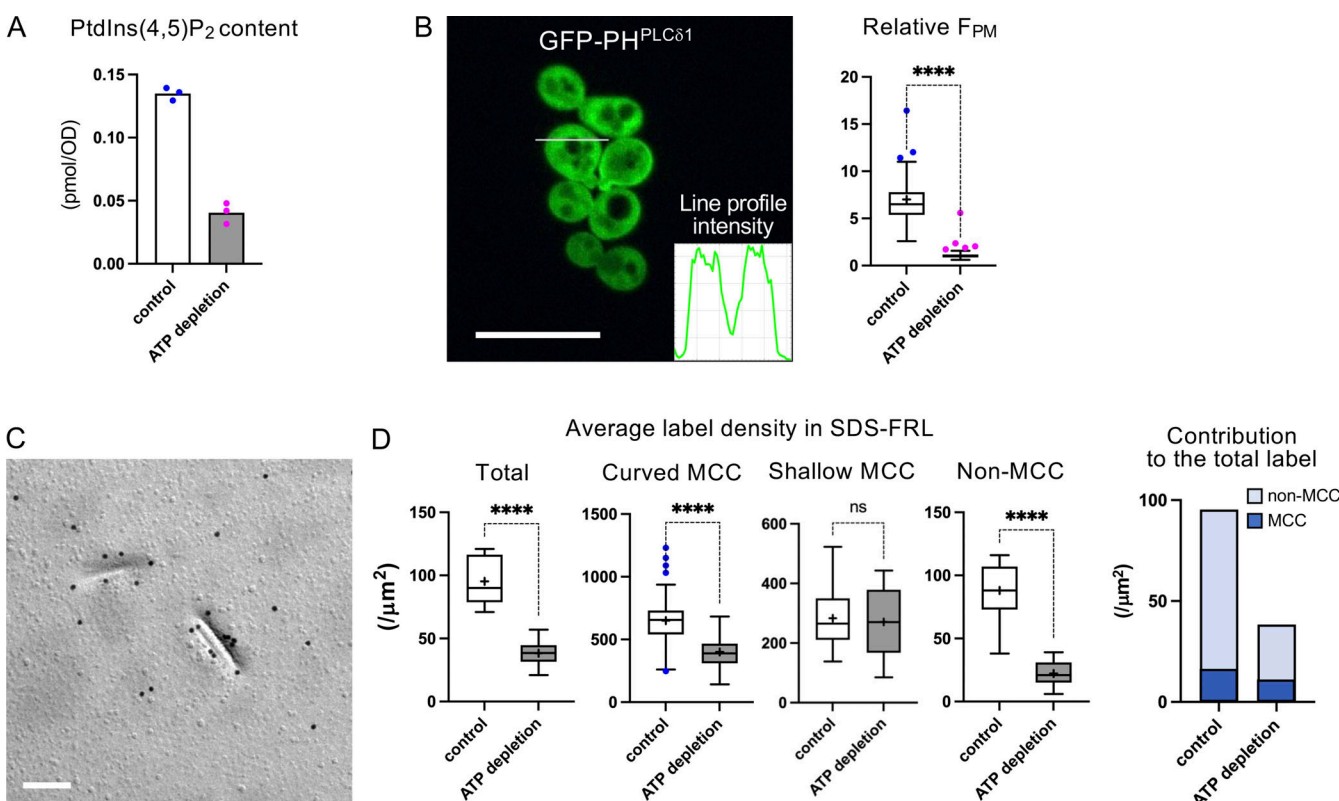

Figure 2. **PtdIns(4,5)P₂ in yeast after acute ATP depletion. (A)** Total cellular content of PtdIns(4,5)P₂ quantified by PRMC-MS ($n = 3$). **(B)** PtdIns(4,5)P₂ shown by fluorescence biosensor GFP-PH$^{PLCδ1}$. Bar, 10 μm. The relative $F_{PM}$ in the control and after ATP depletion. Mann–Whitney test ($n = 42$ [control], 41 [ATP depletion]). ****$P < 0.0001$. **(C)** SDS-FRL of PtdIns(4,5)P₂. Bar, 0.2 μm. **(D)** The label intensity by SDS-FRL in the control and after ATP depletion. Mann–Whitney test (ATP depletion: $n = 10$ [total cell], 58 [curved MCC], 16 [shallow MCC], and 15 [non-MCC]). ****$P < 0.0001$. The bar graph shows the proportion of labels in MCC/eisosomes and non-MCC/eisosome areas.

whereas fluorescence biosensor GFP-PH$^{PLCδ1}$ fails to detect PtdIns(4,5)P₂ even when a significant amount exists.

**PtdIns(4,5)P₂ distribution in PC12 plasma membrane**

The two-dimensional distribution of membrane lipids in adherent cells has often been studied using unroofed membrane preparations. In this method, cells cultured on a flat substrate were mechanically disrupted to remove the dorsal plasma membrane and the cytoplasm, and the basal plasma membrane left on the substrate, or membrane sheet, is subjected to labeling (Fig. S1 B). Although details vary, in most studies, cells are kept cool, fixed with aldehydes after disruption, and incubated with a lipid-binding probe. By this method, fluorescence labels of PtdIns(4,5)P₂ in the basal plasma membrane of PC12 cells were observed as distinct puncta (Fig. 4 A-(a)) (Aoyagi et al., 2005; van den Bogaart et al., 2011; Wang and Richards, 2012).

In contrast, when the basal plasma membrane of quick-frozen PC12 cells in SDS-treated freeze-fracture replicas was labeled by the same set of probes and observed by fluorescence microscopy, the PtdIns(4,5)P₂ label showed even distribution without any local concentration (Fig. 4 A-(b)). Correspondingly, gold nanoparticle labels for PtdIns(4,5)P₂ were observed randomly by electron microscopy, which was verified by statistical analysis using Ripley's K-function (Fig. 4, B and C). Notably, when PC12 cells were fixed by formaldehyde before quick-

freezing, fluorescence labeling of PtdIns(4,5)P₂ in the freeze-fracture replica showed a mottled pattern, indicating a low degree of clustering (Fig. 4 A-(c)). The clustering was also observed in the gold nanoparticle distribution in electron microscopy (Fig. 4 B) and confirmed by Ripley's K-function analysis (Fig. 4 C).

In the formaldehyde-fixed PC12 cell sample, the PtdIns(4,5)P₂ label density was significantly lower than in the quick-frozen one (Fig. S3 A). We hypothesized that this decrease in labeling occurs because proteins crosslinked by formaldehyde are resistant to the SDS treatment and obstruct the access of labeling probes to PtdIns(4,5)P₂. Supporting this idea, digestion of the SDS-treated freeze-fracture replica with proteinase K significantly increased the PtdIns(4,5)P₂ label density in the fixed sample (Fig. S3 A) and the distribution of the PtdIns(4,5)P₂ label no longer showed clustering (Fig. S3, B and C). This result may seem to suggest that, in the formaldehyde-fixed sample, proper labeling of PtdIns(4,5)P₂ does not occur due to crosslinked proteins, whereas the native distribution of PtdIns(4,5)P₂ per se is preserved. To interrogate this latter point, we examined the effect of fixation in yeast cells, which show differential distribution of PtdIns(4,5)P₂ in MCC/eisosomes and non-MCC areas (Fig. 1). The decrease in the PtdIns(4,5)P₂ label density due to formaldehyde fixation and its restoration by proteinase K treatment were observed in non-MCC areas, as in the PC12 cell

Figure 3. **PtdIns(4,5)P₂ in the stationary phase. (A)** Total cellular content of PtdIns(4,5)P₂ quantified by PRMC-MS ($n$ = 3). **(B)** PtdIns(4,5)P₂ shown by fluorescence biosensor GFP-PH$^{PLCδ1}$. Bar, 10 μm. The relative $F_{PM}$ in the control and stationary phase. Mann–Whitney test ($n$ = 42 [control], 41 [stationary phase]). ****$P < 0.0001$. **(C)** SDS-FRL of PtdIns(4,5)P₂. Bar, 0.2 μm. **(D)** The label intensity by SDS-FRL in the control and stationary phase. Mann–Whitney test (stationary phase: $n$ = 10 [total cell], 78 [curved MCC], 30 [shallow MCC], 15 [non-MCC]). ****$P < 0.0001$. The bar graph shows the proportion of labels in MCC/eisosomes and non-MCC/eisosome areas.

sample (Fig. S3 D). Notably, however, the accumulation of the PtdIns(4,5)P₂ labels in MCC/eisosomes was not observed in the fixed sample, even after the proteinase K treatment (Fig. S3 E). This finding suggests that formaldehyde fixation affects the PtdIns(4,5)P₂ labeling not only by obstructing the probe access but also by changing the distribution of PtdIns(4,5)P₂ itself.

Altogether, the results indicate that PtdIns(4,5)P₂ in live PC12 cells is randomly distributed, while the clustering of the PtdIns(4,5)P₂ labels observed in the membrane sheet method is attributed to the labeling procedure including formaldehyde fixation.

Direct comparison with the actual PtdIns(4,5)P₂ content revealed that GFP-PH$^{PLCδ1}$ fails to detect a significant amount of PtdIns(4,5)P₂ in yeast in ATP-deficient conditions. It has been suspected that fluorescence biosensors do not bind to PtdIns(4,5)P₂ in MCC/eisosome efficiently (Fröhlich et al., 2014), but the present result revealed that it scarcely recognizes PtdIns(4,5)P₂ in non-MCC areas as well after ATP depletion. Poor binding to PtdIns(4,5)P₂ in MCC/eisosome is thought to be caused by the dense protein assembly of the eisosome, preventing the biosensor's access to the membrane. A similar problem is likely to occur in the caveolae of mammalian cells, where PtdIns(4,5)P₂ enrichment is observed by SDS-FRL but not by the fluorescence biosensor method (Fujita et al., 2009). On the other hand, it is not clear why the fluorescence biosensor is insensitive to PtdIns(4,5)P₂ in the non-MCC area. Fluorescence biosensors are generally thought to bind to a free pool of target lipids, which are in equilibrium with a protein-bound pool. Thus, one possible

explanation for the insensitivity may be a decrease of free PtdIns(4,5)P₂ caused by an increase of PtdIns(4,5)P₂-binding proteins. Although such a possibility cannot be excluded, we speculate that some change in the plasma membrane property in ATP-deficient conditions, such as derangement of the ionic environment due to ion pump dysfunction, may make the binding between GFP-PH$^{PLCδ1}$ and PtdIns(4,5)P₂ less efficient.

Divergence of SDS-FRL and fluorescence biosensor results also occurs for phosphatidylserine (PtdSer) in the cytoplasmic leaflet of the endoplasmic reticulum (ER) membrane. Two PtdSer-binding protein domains, lactadherin C2 and evectin-2 PH, do not bind to the ER when expressed as a GFP-tagged biosensor in living cells (Uchida et al., 2011; Yeung et al., 2008), whereas evectin-2 PH binds to the cytoplasmic leaflet of the ER when used as a GST-tagged recombinant protein in SDS-FRL (Tsuji et al., 2019a). This discrepancy also indicates that the binding of lipid-binding domains to their target lipids in living cells may be affected by unidentified factors, whereas SDS-FRL is immune from such factors because it uses a membrane preparation stripped of proteins, and the labeling is carried out in a defined condition. What factors are involved in changing the affinity between lipid-binding domains and their target lipids remains to be studied.

This study also suggests that the clustering of the PtdIns(4,5)P₂ labels in formaldehyde-fixed PC12 cells, as observed using fluorescence labeling (Aoyagi et al., 2005; van den Bogaart et al., 2011; Wang and Richards, 2012), may result partially from proteins crosslinked by the fixative obstructing the access of

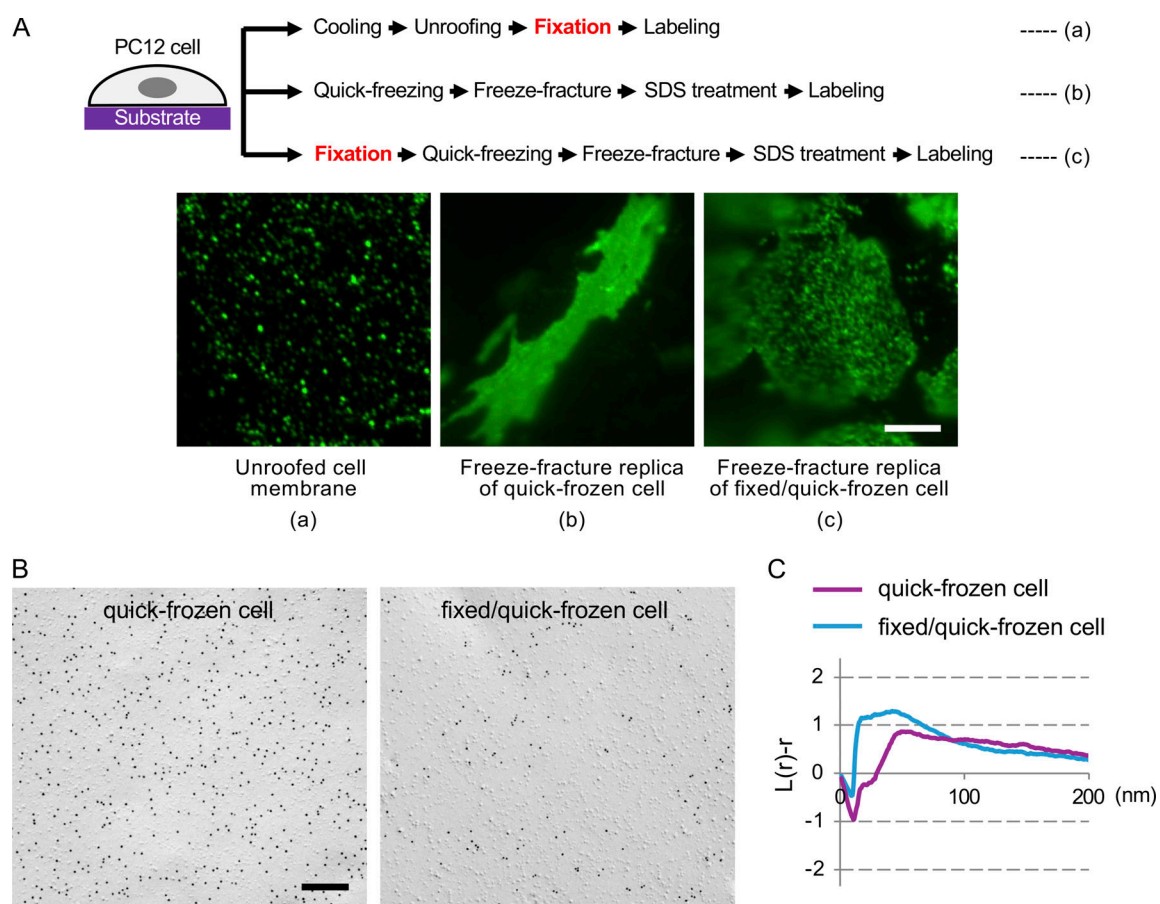

**Figure 4. PtdIns(4,5)P₂ in PC12 cells. (A)** Fluorescence labeling of PtdIns(4,5)P₂ in the basal plasma membrane of PC12 cells. **(a)** Unroofed cell membrane, **(b)** SDS-treated freeze-fracture replica of quick-frozen cell, **(c)** SDS-treated freeze-fracture replica of cells fixed before quick-freezing. Bar, 0.5 μm. **(B)** SDS-FRL of PtdIns(4,5)P₂ in the basal plasma membrane of PC12 cells. Bar, 0.2 μm. **(C)** Point pattern analysis of the PtdIns(4,5)P₂ label distribution in SDS-FRL by Ripley's K-function ($n$ = 10). The result is presented in the normalized form or L-function.

labeling probes to PtdIns(4,5)P₂. On the other hand, the absence of PtdIns(4,5)P₂ label accumulation in yeast MCC/eisosomes in fixed samples, even post-proteinase K treatment, suggests that formaldehyde fixation alters the distribution of PtdIns(4,5)P₂ itself. Given that PtdIns(4,5)P₂ does not have functional groups reactive with aldehydes, formaldehyde is unlikely to act directly on PtdIns(4,5)P₂. We speculate that the formaldehyde fixation affects the distribution of proteins and amino-containing lipids, mainly PtdSer and phosphatidylethanolamine, thereby leading to secondary redistribution of PtdIns(4,5)P₂.

The clustering observed by SDS-FRL of chemically fixed cells is of a lower degree than that observed by unroofed membrane labeling, but other procedures of the latter method may also influence PtdIns(4,5)P₂ distribution. Cooling and mechanical disruption may affect lipid distribution through phase separation and membrane tension alteration, respectively (Fujita et al., 2007; Riggi et al., 2018). Because lipids remain diffusible even after fixation (Tanaka et al., 2010), the binding of probes may further induce the redistribution of target lipids.

Altogether, the present study demonstrated that SDS-FRL can define endogenous PtdIns(4,5)P₂ distribution reliably and in a semiquantitative manner, verifying that its theoretical advantage is implemented in concrete examples. Naturally, SDS-FRL

has its own limitations (Takatori et al., 2014). With regards to the yeast study, deep membrane invaginations, such as curved MCC/eisosome, may not be fractured entirely so that a portion of them is not retained in freeze-fracture replicas (Fig. 1 A). This can lead to an underestimation of the number of labels in the curved MCC/eisosome. Another problem is related to the measurement of the label density. Because two-dimensional micrographs are used for quantification, the area of curved or oblique membranes can be underestimated, which in turn can cause an overestimation of the label density in MCC/eisosomes. For the PC12 study, the effect of unroofing and probe binding on the PtdIns(4,5)P₂ distribution could not be studied because freeze-fracture images cannot distinguish membranes that are unroofed successfully from those that are not. Although these limitations must be taken into consideration, we think that SDS-FRL should be added to the toolbox whenever membrane lipid distribution needs to be studied in detail.

## Materials and methods
### Yeast
All yeast strains used in this study are based on the parent strain SEY6210 (Table S1). They were cultured in YPD medium (1%

yeast extract, 2% polypeptone, and 2% glucose) or in synthetic complete (SC) medium (0.67% yeast nitrogen base without amino acids and with ammonium sulfate [Becton Dickinson], 2% glucose, 0.1255% dropout mix) and used at $OD_{600\,nm} \approx 0.5$. For acute ATP depletion, 2% 2-deoxyglucose and 3 mM sodium azide were added to the culture medium. To induce the stationary phase, cells were cultured in SC medium for 3 days starting at $OD_{600\,nm} \approx 0.15$ (Tsuji et al., 2017).

## PC12 cell

PC12 cells (RRID:CVCL_0481) were obtained from the RIKEN BRC Cell Bank through the National Bio-Research Project of the MEXT/AMED, Tokyo, Japan, and cultured in Dulbecco's modified Eagle's medium supplemented with 10% fetal calf serum and antibiotics in a humidified chamber containing 5% $CO_2$.

## Probes

Recombinant GST-PLCδ1 PH domain (GST-PH$^{PLCδ1}$) and its mutated version, GST- PH$^{PLCδ1(K30N,K32N)}$, were prepared as described (Fujita et al., 2009). GFP-PH$^{PLCδ1}$ was prepared by treating GST-GFP-PH$^{PLCδ1}$ with GST-fused HRV 3 C protease for 1.5 h on ice and then removing excised GST and the protease by a glutathione resin column. Rabbit anti-GFP antibody was donated by Dr. Masahiko Watanabe (Hokkaido University, Sapporo, Japan). Rabbit anti-GST antibody (A190-122A, RRID:AB_67419; Bethyl), protein A conjugated to 10 nm colloidal gold (PAG10, The University Medical Center Utrecht, Utrecht, Netherlands), and Alexa488-conjugated donkey anti-rabbit IgG (711-545-152, RRID: AB_2313584; Jackson ImmunoResearch Lab) antibody were obtained from respective suppliers.

## Quick-freezing and freeze-fracture

For yeast, a copper EM grid (200 mesh; Nisshin EM) was immersed in a cell pellet, sandwiched between a flat aluminum disc (Engineering Office M. Wohlwend) and a thin copper foil (20 μm thick; Nilaco), and frozen using HPM 010 or HPM100 high-pressure freezing machines (Leica) according to the manufacturer's instruction. For PC12, cells cultured on a 20-μm-thick gold foil were placed on the flat aluminum disc with the cell side down and quick-frozen by HPM010. For some specimens, yeast and PC12 cells were fixed with 4% formaldehyde in 0.1 M phosphate buffer for 30 min before quick-freezing. The frozen specimens were transferred to a cold stage of a BAF 400 (Balzers) or an ACE900 (Leica) apparatus and freeze-fractured at −115 to −105°C under a vacuum of ∼$1 \times 10^{-6}$ mbar. Replicas were produced by the electron-beam evaporation of carbon (2–5 nm thick), followed by platinum/carbon (Pt/C) (2 nm thick), and then by carbon (20 nm thick) as described previously (Fujita et al., 2010).

Thawed replicas were treated with 2.5% SDS in 0.1 M Tris-HCl (pH 8.0) at 60°C overnight. The yeast cell walls were removed by treating the replicas for 2 h at 30°C with 0.5% Westase (Takara Bio) in McIlvain citrate-phosphate buffer (pH 6.0) containing 10 mM EDTA, 30% FCS, and a protease inhibitor cocktail (Nacalai), followed by treatment at 60°C overnight with 2.5% SDS. Replicas for protein labeling were used as they were, whereas those for phospholipid labeling were further treated at 37°C overnight with 50 μg/ml proteinase K (TAKARA) in 20 mM Tris-HCl (pH8.0), 10 mM EDTA, 10 mM NaCl, and 0.5% SDS. The replicas were rinsed extensively with 2.5% SDS in 0.1 M Tris-HCl (pH 8.0) and then phosphate-buffered saline (PBS) containing 0.1% Tween-20 (PBST) before labeling.

## SDS-FRL for electron microscopy

Freeze-fracture replicas were treated with 3% bovine serum albumin (BSA) in PBS for blocking and incubated at 4°C overnight with GST-PH$^{PLCδ1}$ (62.5 or 12.5 ng/ml) in 1% BSA in PBS (Fujita et al., 2009). They were further incubated at 37°C for 30 min with rabbit anti-GST antibody (5 μg/ml) and with 50-fold-diluted colloidal gold (10 nm)-conjugated Protein A in 1% BSA in PBS. For labeling of Sur7-GFP and Pma1-GFP, replicas were blocked with 3% BSA in PBS and incubated at 4°C overnight with anti-GFP antibody (10 μg/ml) followed by colloidal gold-conjugated Protein A in 1% BSA in PBS at 37°C for 1 h. The labeled replicas were picked up on formvar-coated EM grids and observed with a JEM-1011 electron microscope (JEOL) operated at 100 kV.

The labeling density (the number of colloidal gold particles per unit area) was obtained by counting the number of colloidal gold particles manually and measuring the area using ImageJ (NIH; RRID:SCR_003070). For yeast non-MCC areas and PC12 cells, square areas of $1 \times 1$ μm were selected randomly, whereas, for yeast MCC/eisosomes, the smallest rectangles covering the furrows were measured.

## Point pattern analysis of SDS-FRL label

Gold labels in areas of $1 \times 1$ μm chosen randomly were analyzed by Ripley's K-function (Fujita et al., 2007; Ripley, 1979). For significance tests, 99% confidence envelopes for complete spatial randomness (CSR) were generated from 100 Monte Carlo simulations. The graphs in the result are presented in the normalized format or the L-function.

## Fluorescence microscopy of yeast expressing GFP-PH$^{PLCδ1}$

Images of yeast expressing GFP-PH$^{PLCδ1}$ were taken by an Axiovert 200 M microscope equipped with Apotome2 (Carl Zeiss) using an Apochromat 63x or 100x objective lens and the acquisition software ZEN (blue edition, Carl Zeiss) and processed using Fiji/ImageJ (RRID:SCR_002285) and Adobe Photoshop CC2018 (RRID:SCR_014199). Plasma membrane relative fluorescence (relative $F_{PM}$) was obtained as described before (Nishimura et al., 2019), that is, the fluorescence intensity of the plasma membrane ($F_{cross}$) was obtained by averaging two peaks on a line crossing over a cell. Using $F_{cross}$ and the average values of fluorescence intensity inside and outside a cell ($F_{in}$ and $F_{out}$, respectively), relative $F_{PM}$ was calculated by the equation: relative $F_{PM} = (F_{cross}-F_{out})/(F_{in}-F_{out})$. An Excel VBA macro for automatic calculations was provided by Dr. Taki Nishimura (University College London, London, UK).

## Fluorescence microscopy of the unroofed membrane preparation

PC12 cells cultured on glass coverslips were cooled on ice and the dorsal plasma membrane was disrupted by pressing and

removing a nitrocellulose membrane. The obtained membrane sheet was fixed for 15 min with 4% formaldehyde in 0.1 M phosphate buffer, treated with 3% BSA in PBS for blocking, and incubated at 4°C overnight with GST-PH$^{PLC\delta1}$ in 1% BSA in PBS. The samples were further treated with rabbit anti-GST antibody (RRID: AB_67419) and Alexa488-conjugated donkey anti-rabbit IgG antibody (RRID:AB_67419) and observed by fluorescence microscope.

### SDS-FRL for fluorescence microscopy
Freeze-fracture replicas were blocked and labeled with GST-PH$^{PLC\delta1}$, rabbit anti-GST antibody, and Alexa488-conjugated donkey anti-rabbit IgG antibody in the same manner as the unroofed membrane preparation. The replicas were overlaid with a coverslip for fluorescence microscopy.

### Liposome experiments
Liposomes were prepared as described previously (Tsuji et al., 2019a) with minor modifications. Dioleoyl phosphatidylcholine (850375P; Avanti Polar Lipids), 18:1 PtdIns(4,5)P$_2$ (850155P; Avanti Polar Lipids), and 16:0 Liss Rhod phosphatidylethanolamine (810158P; Avanti Polar Lipids) were mixed in a glass vial at a molar ratio of 94:5:1 and dried under nitrogen gas. The resulting lipid film was vacuum-dried overnight, rehydrated with 20 mM HEPES-NaOH (pH 7.4), and extruded 11 times through a 400-nm pore-size membrane. Subsequently, liposomes were incubated with 0.5 mg/ml of GFP-PH for 5 min at 37°C and subjected to quick-freezing for freeze-fracture replica preparation or fluorescence microscopy using FV3000RS confocal laser scanning microscope (Olympus) equipped with a 100x objective lens (Olympus).

### Quantification of PtdIns(4,5)P$_2$ by PRMC-MS
Yeast was centrifuged for 1 min at 4,000 rpm after measuring OD$_{600}$ and the pellet was quickly frozen. The frozen pellet was disrupted by vortexing for 10 min with chloroform/methanol (2:1) and 0.5 mm Zirconia/Silica Beads frozen yeast pellet. The supernatant of the mixture was further extracted with chloroform and 50 mM citrate, and the lipid layer was dried and subjected to PRMC-MS to measure PtdIns(4,5)P$_2$ (Morioka et al., 2022). Briefly, lipids mixed with internal standards were applied to diethylaminoethyl–cellulose column chromatography to condense anionic phospholipids, methylated with trimethylsilyl diazomethane, separated on a Chiral high performance liquid chromatography column (DAICEL), and subjected to electrospray ionization-tandem mass spectrometry. The PtdIns(4,5)P$_2$ peak area was normalized to the internal/surrogate standard.

### Statistical analysis
Statistical differences between samples were examined by the Mann–Whitney test. In the box plots, the center lines show the median, box boundaries indicate the 25th and 75th percentiles, and whiskers are Tukey-type. Statistical analyses were performed with Prism 8 (RRID:SCR_002798; GraphPad).

### Online supplemental material
Fig. S1 illustrates the methods employed in this study. Fig. S2 shows the PtdIns(4,5)P$_2$ labeling in yeast and the result of the liposome assay. Fig. S3 shows the PtdIns(4,5)P$_2$ labeling in PC 12 cells under different experimental conditions. Table S1 shows yeast strains used in this study.

### Data availability
All the data and relevant materials that support the findings of this study are available from the corresponding author upon reasonable request.

## Acknowledgments
We thank Dr. Taki Nishimura (University College London) for the macro programs and advice on fluorescence quantification, Dr. Masahiko Watanabe (Hokkaido University) for anti-GFP antibody, and Dr. Jinglei Cheng (Nagoya University), Dr. Konomi Marumo, Ms. Ayaka Saito (Hokkaido University), and Ms. Hiroko Osakada-Iwamoto (Juntendo University) for excellent technical assistance.

This study was supported by Grants-in-Aid for Scientific Research from the Japan Society of the Promotion of Science to T. Tsuji (19K07265) and T. Fujimoto (18H04023 and 22H00446), by grants from Takeda Medical Science Foundation and Nakatani Foundation for Advancement of Measuring Technologies in Biomedical Engineering to T. Fujimoto, and by Nanken-Kyoten, TMDU, to J. Hasegawa and T. Sasaki. Open Access funding provided by Juntendo University.

Author contributions: T. Tsuji: Data curation, Formal analysis, Funding acquisition, Investigation, Resources, Validation, Visualization, Writing - original draft, Writing - review & editing, J. Hasegawa: Investigation, Writing - review & editing, T. Sasaki: Formal analysis, Investigation, Methodology, Project administration, Resources, Validation, Writing - original draft, T. Fujimoto: Conceptualization, Data curation, Formal analysis, Funding acquisition, Investigation, Methodology, Project administration, Resources, Supervision, Validation, Visualization, Writing - original draft, Writing - review & editing.

Disclosures: The authors declare no competing interests exist.

Submitted: 14 November 2023

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

## A Fluorescence biosensor method

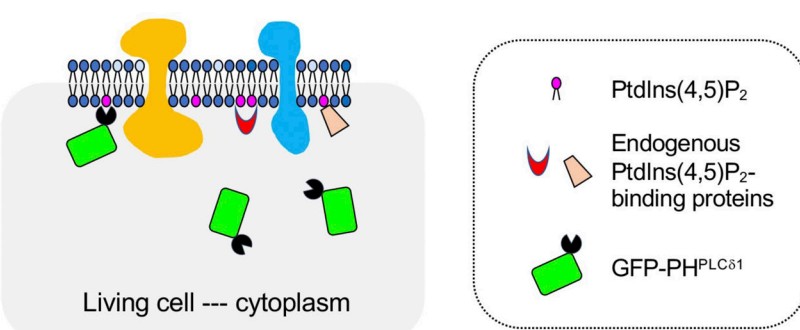

## B Unroofed membrane method

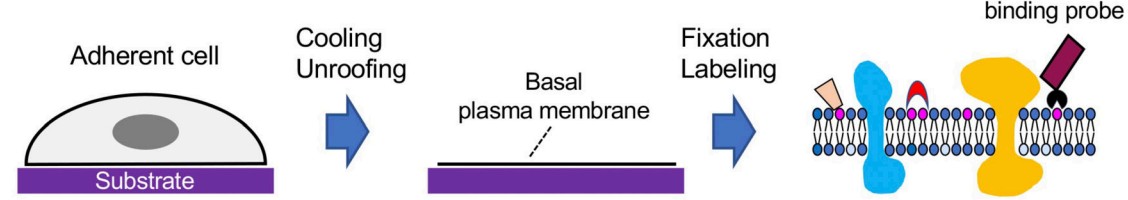

## C SDS-FRL of quick-frozen cells

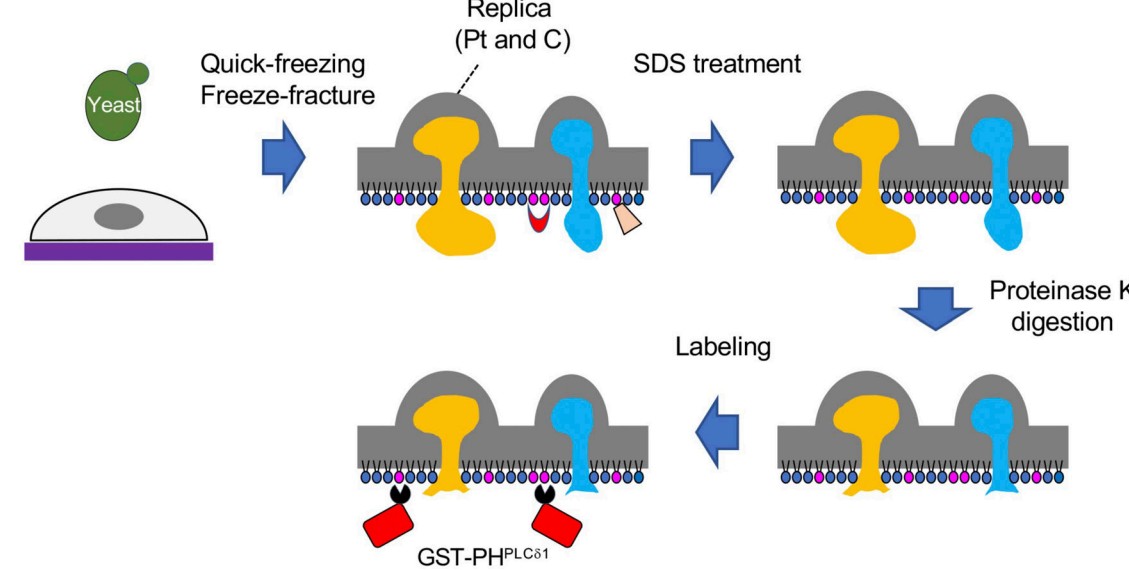

Figure S1. **Diagrams of methods used in this study. (A)** Fluorescence biosensor method. GFP-PH$^{PLC\delta1}$ expressed in living cells binds to PtdIns(4,5)P$_2$ in cellular membranes. **(B)** Unroofed membrane method. Culture cells on a substrate are stripped of the dorsal plasma membrane and the cytoplasm, and the basal plasma membrane remaining on the substrate is chemically fixed and labeled for PtdIns(4,5)P$_2$. **(C)** SDS-FRL of quick-frozen cells. Cells are quick-frozen and freeze-fractured to prepare replicas. The membrane physically stabilized by the platinum and carbon coating is labeled for PtdIns(4,5)P$_2$ after SDS treatment. In the present study, some of the replicas before labeling are further treated with protease K to digest integral membrane proteins.

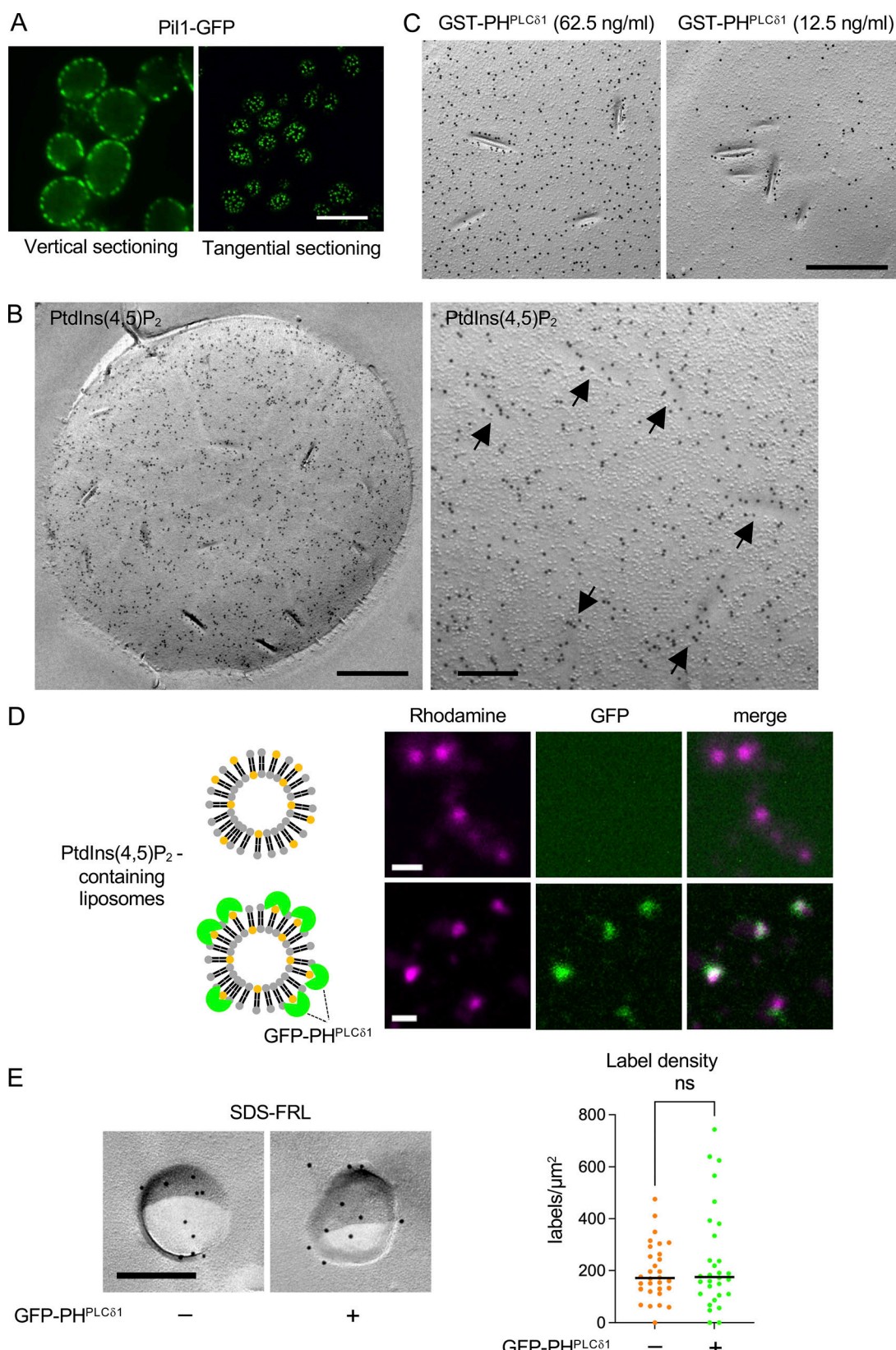

Figure S2. **The PtdIns(4,5)P₂ labeling in yeast and liposomes. (A)** Distribution of Pil1-GFP observed by conventional fluorescence microscopy. Bar, 10 µm. **(B)** PtdIns(4,5)P₂ labeling by SDS-FRL. MCC (arrows). Bars, 0.5 mm (left), 0.2 µm (right). **(C)** PtdIns(4,5)P₂ labeled by two different concentrations of GST-PH^PLCδ1. Bar, 0.5 µm. **(D)** Liposomes containing 5% PtdIns(4,5)P₂ and 1% Liss Rhod phosphatidylethanolamine incubated with or without GFP-PH^PLCδ1. Bar, 1 µm. **(E)** SDS-FRL of PtdIns(4,5)P₂-containing liposomes with or without GFP-PH^PLCδ1. Bar, 0.2 µm. The PtdIns(4,5)P₂ label density in the liposome by SDS-FRL. Mann–Whitney test ($n$ = 27 [without GFP-PH^PLCδ1], 25 [with GFP-PH^PLCδ1]).

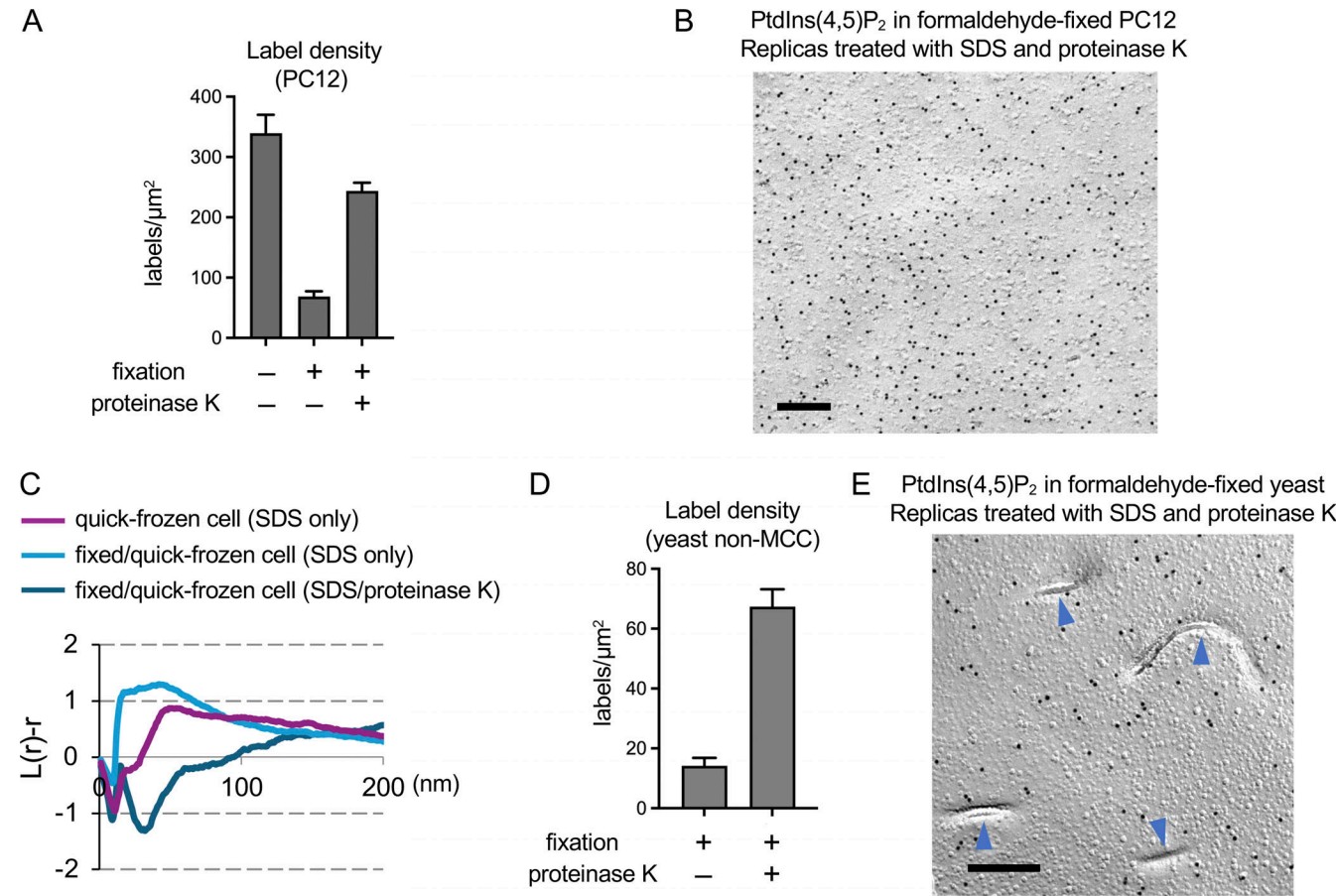

**A** Label density (PC12)

**B** PtdIns(4,5)P2 in formaldehyde-fixed PC12 Replicas treated with SDS and proteinase K

**C**
- quick-frozen cell (SDS only)
- fixed/quick-frozen cell (SDS only)
- fixed/quick-frozen cell (SDS/proteinase K)

**D** Label density (yeast non-MCC)

**E** PtdIns(4,5)P2 in formaldehyde-fixed yeast Replicas treated with SDS and proteinase K

Figure S3. **The PtdIns(4,5)₂ labeling in PC12 cells. (A)** PtdIns(4,5)P2 label density in PC12 cell samples. Freeze-fracture replicas of formaldehyde-fixed PC12 cells were treated with SDS alone or with SDS and proteinase K before labeling. **(B)** PtdIns(4,5)$P_2$ labeling in formaldehyde-fixed PC12 cells. Freeze-fracture replicas were treated with SDS and proteinase K. Bar, 0.2 μm. **(C)** Point pattern analysis of the PtdIns(4,5)$P_2$ label distribution by L-function ($n$ = 10). The result of formaldehyde-fixed PC12 cells by using SDS/proteinase K-treated freeze-fracture replicas was added to Fig. 4 C. **(D)** PtdIns(4,5)$P_2$ label density in non-MCC areas of formaldehyde-fixed budding yeast. Comparison between freeze-fracture replicas treated with SDS alone or with SDS and proteinase K. **(E)** PtdIns(4,5)$P_2$ labeling in formaldehyde-fixed budding yeast. Freeze-fracture replicas were treated with SDS and proteinase K. MCC/eisosomes (arrowheads). Bar, 0.2 μm.

**Provided online is Table S1. Table S1 shows yeast strains used in this study.**

