## [Peer Review File · The Journal of Cell Biology]

Definition of phosphatidylinositol 4,5-bisphosphate distribution by freeze-fracture replica labeling

Takuma Tsuji, Junya Hasegawa, Takehiko Sasaki, and Toyoshi Fujimoto

Corresponding Author(s): Toyoshi Fujimoto, Juntendo University

Review Timeline:

Submission Date:	2023-11-14
Editorial Decision:	2024-01-09
Revision Received:	2024-09-18
Editorial Decision:	2024-10-09
Revision Received:	2024-10-15

Monitoring Editor: Lois Weisman

Scientific Editor: Andrea Marat

Transaction Report:

DOI: <https://doi.org/10.1083/jcb.202311067>

January 9, 2024

Re: JCB manuscript #202311067

Prof. Toyoshi Fujimoto
Juntendo University
Graduate School of Medicine
2-1-1 Hongo, Bunkyo
Tokyo, Tokyo 113-8421
Japan

Dear Prof. Fujimoto,

Thank you for submitting your manuscript entitled "Definition of phosphatidylinositol 4,5-bisphosphate distribution by freeze-fracture replica labeling". First our apologies for the delay in providing you with this decision and thanks for your patience. The manuscript has been evaluated by expert reviewers, whose reports are appended below. Unfortunately, after an assessment of the reviewer feedback, our editorial decision is against publication in JCB.

As you will see, both reviewers agree that the field needs a better way to detect the full population of phosphatidylinositol 4,5-bisphosphate at the plasma membrane, but each reviewer is not yet convinced that this is achieved by your new protocol. Reviewer 1 focused on the need for additional analysis and quantitation of your data. For example, for Figure 1E, the reviewer suggests that for each type of region, you might measure the distance between each gold particle and present the values as a density distribution. Also, in Figure 4, they point out that in addition to a change in distribution of phosphatidylinositol 4,5-bisphosphate, it appears that there is also much less PtdIns(4,5)P₂. They suggest that you quantitate this change.

The other critical issue that was raised by both reviewers was a more thorough study of the impact of SDS treatment. Since this is new to the field, you will need additional approaches to test the assumption that all of the PtdIns(4,5)P₂ is being immobilized by the SDS-FRL method.

Although your manuscript is intriguing, I feel that the points raised by the reviewers are more substantial than can be addressed in a typical revision period. If you wish to expedite publication of the current data, it may be best to pursue publication at another journal.

Given interest in the topic, I would be open to resubmission to JCB of a significantly revised and extended manuscript that fully addresses all the reviewers' concerns and is subject to further peer-review. If you would like to resubmit this work to JCB, please contact the journal office to discuss an appeal of this decision or you may submit an appeal directly through our manuscript submission system. Please note that priority and novelty would be reassessed at resubmission.

Regardless of how you choose to proceed, we hope that the comments below will prove constructive as your work progresses. We would be happy to discuss the reviewer comments further once you've had a chance to consider the points raised in this letter. You can contact the journal office with any questions at cellbio@rockefeller.edu.

Thank you for thinking of JCB as an appropriate place to publish your work.

Sincerely,

Lois Weisman, PhD
Monitoring Editor

Andrea L. Marat, PhD
Senior Scientific Editor

Journal of Cell Biology

Reviewer #1 (Comments to the Authors (Required)):

The native unperturbed distribution of lipids in the plasma membrane has been challenging to visualize. Classically, fluorescent biosensors have been used to map different lipids in live and fixed cells. Electron microscopy has also been used to study lipids at the nanoscale in both frozen and fixed cells. Many discrepancies and controversies exist between the various studies that

have attempted to map the distribution of specific lipids. Thus, in this work by Tsuji et al. the distribution of PIP2 in the plasma membrane of yeast and mammalian cells is explored by quantitative mass spec, fluorescence, and freeze fracture immune-gold platinum replica electron microscopy. The goal of the study was to see if specific methods better preserve the native "live-cell" "real" distribution of lipids. I find this relatively short paper interesting. It is an important (but thorny) issue in cell biology. I have specific comments that the authors might consider to improve the manuscript. Most importantly, I believe the paper would benefit from much more quantitative/statistical analysis of the images in the paper to solidify their findings.

1. The quantitation of gold labeling in Figure 1E is difficult to judge. The numbers should be presented in terms of particles/area and the error presented across different cells or membranes. As it is presented now, I can not evaluate the robustness of these data. Please include more details about how this analysis was done. For example, might the authors measure the distance of each gold particle to the uncurved and curved MIC and present the numbers as a density distribution vs distance?
2. It would be helpful if the authors explain why the SDS is important in the labeling reaction.
3. The fluorescence panels in figure 4 appear to be mislabeled. Where are the (i) panels? Also, I don't think the scale bar could possibly be 0.5 microns for these images.
4. The amount of PIP2 in figure 4 looks sustainably depleted in the fixed/quick-frozen sample. Is the change in distribution an effect of this depletion? Please quantify the amount of PIP2 in the membrane of these two or three samples.

Reviewer #2 (Comments to the Authors (Required)):

The manuscript by Tsuji et al describes the use of freeze-fracture replicas to study the distribution of phosphatidylinositol 4,5-bisphosphate in biological membranes. The authors use recombinant GST-PLC 1 PH domain in combination with anti-GST antibodies conjugated to colloidal gold to visualize the phosphoinositide in replicas that have been extracted with SDS and then treated with proteases to remove any proteins obstructing access to the probe. After analyzing both yeast and mammalian samples and comparing it with fluorescence determinations in live cells, the authors conclude that "SDS-FRL should be added to the toolbox whenever membrane lipid distribution needs to be studied in detail."

The use of freeze fracture replicas in combination with recombinant GST-PLC 1 PH domain to study the distribution of phosphatidylinositol 4,5-bisphosphate is not entirely novel. Indeed, the same group introduced this approach more than 10 years ago (Fujita et al, 2009). The use of SDS and proteases employed in the present manuscript (introduced earlier in studies of phosphatidylserine distribution) seemingly improves access to lipid molecules associated with or obscured by proteins.

The use of SDS assumes that lipids, including phosphatidylinositol 4,5-bisphosphate, are perfectly immobilized by the deposition of the carbon and platinum layer, preventing them from being extracted or redistributed upon treatment with the detergent or proteases. I could not find any data or references in the current manuscript to validate this important assumption. Without convincing evidence to this effect, the significance of the data becomes questionable.

The second concern is whether the lipids detected by the SDS-FRL method are representative of the overall phosphatidylinositol 4,5-bisphosphate population, or reflect preferentially lipids that are tightly bound to proteins and are therefore poorly exchangeable with the bulk lipids, accounting for the fraction of phosphatidylinositol 4,5-bisphosphate that is preserved after ATP depletion. This is suggested by the lower fractional change in the MCC-associated phosphatidylinositol 4,5-bisphosphate compared to non-MCC phosphatidylinositol 4,5-bisphosphate following ATP depletion (Figure 2C). If this interpretation is correct, the selective properties of the method (i.e., preferential detection of protein-occluded, non-exchangeable phosphatidylinositol 4,5-bisphosphate) should be stressed.

Point-by-point response

Reviewer #1:

The native unperturbed distribution of lipids in the plasma membrane has been challenging to visualize. Classically, fluorescent biosensors have been used to map different lipids in live and fixed cells. Electron microscopy has also been used to study lipids at the nanoscale in both frozen and fixed cells. Many discrepancies and controversies exist between the various studies that have attempted to map the distribution of specific lipids. Thus, in this work by Tsuji et al. the distribution of PIP2 in the plasma membrane of yeast and mammalian cells is explored by quantitative mass spec, fluorescence, and freeze fracture immune-gold platinum replica electron microscopy. The goal of the study was to see if specific methods better preserve the native "live-cell" "real" distribution of lipids. I find this relatively short paper interesting. It is an important (but thorny) issue in cell biology. I have specific comments that the authors might consider to improve the manuscript. Most importantly, I believe the paper would benefit from much more quantitative/statistical analysis of the images in the paper to solidify their findings.

We are grateful to the reviewer for recognizing the importance of the subject matter discussed in our manuscript and for finding it interesting. We really appreciate his/her insightful comments.

1. The quantitation of gold labeling in Figure 1E is difficult to judge. ^{a)} The numbers should be presented in terms of particles/area and ^{b)} the error presented across different cells or membranes. As it is presented now, I cannot evaluate the robustness of these data. ^{c)} Please include more details about how this analysis was done. ^{d)} For example, might the authors measure the distance of each gold particle to the uncurved and curved MIC and present the numbers as a density distribution vs distance?

- a) We apologize for the difficulty that the reviewer found in interpreting the quantitation data. In the revised manuscript, annotations were added to Figure 1E (and other figures) to clearly show that the labeling intensity is measured as the density of immunogold particles in a unit area of freeze-fracture replicas (particles/ μm^2).
- b) To present the reproducibility of the freeze-fracture replica labeling, the labeling intensity across different cells was measured and shown in Supplementary Figure 2E by using SuperPlots (Lord et al, JCB 219, e202001064, 2020). This result should enable readers to evaluate to what extent the labeling intensity in freeze-fracture replicas can be varied in different cells.
- c) The Materials and Methods section was revised to give more details about how the labeling intensity was measured.
- d) We appreciate that the reviewer suggested an alternative way to evaluate how the labels are distributed in relation to MCC/eisosomes. In the revised manuscript, besides the measurement of the label density in a unit area, the distribution of the label was quantified according to the distance from the center line of MCC/eisosomes. The result clearly shows that the labeling intensity is the highest within 0.25 μm from the center line of MCC/eisosomes and decreases as the distance from MCC/eisosomes becomes larger. This result was added as Supplementary Figure 2C of the revised manuscript.

2. It would be helpful if the authors explain why the SDS is important in the labeling reaction.

SDS is essential in the labeling reaction as it effectively removes peripheral membrane proteins and cytoplasmic proteins adhering to freeze-fracture replicas. In the development of the freeze-fracture labeling method, SDS was found to provide specific immunolabeling for integral membrane proteins (Fujimoto, *J Cell Sci* 108, 3443–3449, 1996) and for phospholipids (Fujimoto et al, *J Cell Sci* 109, 2453–2460, 1996). SDS's ability to clean the replica surface is crucial, as other detergents like Triton X-100 do not have comparable cleaning efficacy. We have added the rationale for using SDS was added to the Introduction section of the revised manuscript.

3. The fluorescence panels in figure 4 appear to be mislabeled. Where are the (i) panels? Also, I don't think the scale bar could possibly be 0.5 microns for these images.

We apologize for overlooking the careless mislabeling in Figure 4A. We corrected the figure in the revised manuscript. The scale bar in Figure 4B is correct.

4. The amount of PIP₂ in figure 4 looks sustainably depleted in the fixed/quick-frozen sample. Is the change in distribution an effect of this depletion? Please quantify the amount of PIP₂ in the membrane of these two or three samples.

We appreciate the reviewer for raising this issue. The PI(4,5)P₂ label density in the fixed/quick-frozen samples was indeed significantly lower compared to the quick-frozen samples, which may explain why PI(4,5)P₂ labels show clustering only in the fixed/quick-frozen samples. We hypothesized that this decrease in labeling could be due to proteins crosslinked by formaldehyde, which may resist SDS treatment and remain bound to the replica surface, hindering access of labeling probes to PI(4,5)P₂. Supporting this hypothesis, further treatment of freeze-fracture replicas with proteinase K, which digests proteins, significantly increased the PI(4,5)P₂ label density compared to SDS treatment alone, and eliminated the clustering of PI(4,5)P₂ labels. This result suggests that steric hindrance from crosslinked proteins may be a cause of the apparent clustering of PI(4,5)P₂ labels in formaldehyde-fixed samples.

However, it is important to note that the distribution of PI(4,5)P₂ labels in aldehyde-fixed cells may not reflect their distribution in living cells. This was made clear by fixing yeast by formaldehyde before quick-freezing, and comparing the PI(4,5)P₂ label in the fixed/quick-frozen sample versus the quick-frozen sample. In non-MCC areas of yeast, PI(4,5)P₂ labeling was significantly reduced in the fixed sample, but increased after proteinase K digestion, similar to observations in PC12 cells. In contrast, in MCC/eisosomes where intense labeling is seen in quick-frozen samples, PI(4,5)P₂ labeling remained scarce in fixed/quick-frozen samples even after proteinase K digestion. These findings suggest that formaldehyde fixation alters the distribution of PI(4,5)P₂ itself, and that the label obtained in fixed sample does not represent native PI(4,5)P₂ distribution.

The results described above are incorporated to Supplementary Figure 4 and discussed in the revised manuscript.

Reviewer #2 (Comments to the Authors (Required)):

The manuscript by Tsuji et al describes the use of freeze-fracture replicas to study the distribution of phosphatidylinositol 4,5-bisphosphate in biological membranes. The authors use recombinant GST-PLC δ 1 PH domain in combination with anti-GST antibodies conjugated to colloidal gold to visualize the phosphoinositide in replicas that have been extracted with SDS and then treated with proteases to remove any proteins obstructing access to the probe. After analyzing both yeast and mammalian samples and comparing it with fluorescence determinations in live cells, the authors conclude that "SDS-FRL should be added to the toolbox whenever membrane lipid distribution needs to be studied in detail."

The use of freeze fracture replicas in combination with recombinant GST-PLC δ 1 PH domain to study the distribution of phosphatidylinositol 4,5-bisphosphate is not entirely novel. Indeed, the same group introduced this approach more than 10 years ago (Fujita et al, 2009). The use of SDS and proteases employed in the present manuscript (introduced earlier in studies of phosphatidylserine distribution) seemingly improves access to lipid molecules associated with or obscured by proteins.

The use of SDS assumes that lipids, including phosphatidylinositol 4,5-bisphosphate, are perfectly immobilized by the deposition of the carbon and platinum layer, preventing them from being extracted or redistributed upon treatment with the detergent or proteases. I could not find any data or references in the current manuscript to validate this important assumption. Without convincing evidence to this effect, the significance of the data becomes questionable.

The retention of phospholipids in freeze-fracture replicas is indeed crucial for validating the SDS-FRL method. This issue has been addressed in the following studies, now cited in the manuscript:

1. Fujimoto K, Umeda M, Fujimoto T, Transmembrane phospholipid distribution revealed by freeze-fracture replica labeling, *J Cell Sci* 109, 2453–60, 1996.
2. Fujita A and Fujimoto T, Quantitative retention of membrane lipids in the freeze-fracture replica, *Histochem Cell Biol*, 128, 385-9, 2007.

In these studies, phospholipid retention in freeze-fracture replicas was quantified by measuring inorganic phosphorus, comparing replicas with and without SDS treatment. The result showed that at least 70–80% of phospholipids remain after SDS treatment. Although this might suggest a loss of 20–30% phospholipids, this difference is likely due to overestimation of phospholipids in the control sample (i.e., the replicas without the SDS treatment), which is attributed to the adherence of extraneous liposomes (as discussed in Fujita et al, 2007). While we have not studied the effects of proteinase K treatment, the increased phospholipid labeling density after this treatment indicates that the phospholipid retention in freeze-fracture replicas is unlikely to be compromised by proteinase K.

Additionally, the redistribution of phospholipids within freeze-fracture replicas is considered unlikely due to the solid nature of platinum and carbon layers under the temperatures used during SDS and proteinase K treatments.

These points and references have been incorporated into the revised manuscript.

The second concern is whether the lipids detected by the SDS-FRL method are representative of the overall phosphatidylinositol 4,5-bisphosphate population, or reflect preferentially lipids that are tightly bound to proteins and are therefore poorly exchangeable with the bulk lipids, accounting for the fraction of phosphatidylinositol 4,5-bisphosphate that is preserved after ATP depletion. This is suggested by the lower fractional change in the MCC-associated phosphatidylinositol 4,5-bisphosphate compared to non-MCC phosphatidylinositol 4,5-bisphosphate following ATP depletion (Figure 2C). If this interpretation is correct, the selective properties of the method (i.e., preferential detection of protein-occluded, non-exchangeable phosphatidylinositol 4,5-bisphosphate) should be stressed.

We thank the reviewer for raising this issue. To address the question whether the SDS-FRL method recognizes protein-bound PI(4,5)P₂ preferentially, we conducted a model system study using liposomes. Liposomes containing 5% PI(4,5)P₂ were incubated with or without recombinant GFP-PLCδ1 PH domain (GFP-PH). After confirming the binding of GFP-PLCδ1 PH to the liposomes by fluorescence microscopy, the samples were quick-frozen and subjected to SDS-FRL. The result showed that the labeling intensity of the PI(4,5)P₂ in liposomes preincubated with GFP-PH was not significantly different from that in liposomes without the GFP-PH incubation, indicating that PI(4,5)P₂ is labeled with comparable efficiency regardless of its binding to proteins.

The results of this additional experiment have been incorporated to Supplementary Figure 3 and are discussed in the revised manuscript.

October 9, 2024

RE: JCB Manuscript #202311067R-A

Prof. Toyoshi Fujimoto
Juntendo University
Graduate School of Medicine
2-1-1 Hongo, Bunkyo
Tokyo, Tokyo 113-8421
Japan

Dear Prof. Fujimoto:

Thank you for submitting your revised manuscript entitled "Definition of phosphatidylinositol 4,5-bisphosphate distribution by freeze-fracture replica labeling". We would be happy to publish your paper in JCB pending final revisions necessary to meet our formatting guidelines (see details below).

A. MANUSCRIPT ORGANIZATION AND FORMATTING:

- 1) Text limits: Character count for Reports is < 20,000, not including spaces. Count includes abstract, introduction, * combined results and discussion, and acknowledgments. Count does not include title page, figure legends, materials and methods, references, tables, or supplemental legends.
- 2) Figures limits: Reports may have up to 5 main text figures.
- 3) Figure formatting: Scale bars must be present on all microscopy images, including inset magnifications. Molecular weight or nucleic acid size markers must be included on all gel electrophoresis. Aspect ratios of images may not be altered.
- 4) Statistical analysis: Error bars on graphic representations of numerical data must be clearly described in the figure legend. The number of independent data points (n) represented in a graph must be indicated in the legend. Statistical methods should be explained in full in the materials and methods. For figures presenting pooled data the statistical measure should be defined in the figure legends. Please also be sure to indicate the statistical tests used in each of your experiments (either in the figure legend itself or in a separate methods section) as well as the parameters of the test (for example, if you ran a t-test, please indicate if it was one- or two-sided, etc.). Also, if you used parametric tests, please indicate if the data distribution was tested for normality (and if so, how). If not, you must state something to the effect that "Data distribution was assumed to be normal but this was not formally tested."
- 5) Abstract and title: The abstract should be no longer than 160 words and should communicate the significance of the paper for a general audience. The title should be less than 100 characters including spaces. Make the title concise but accessible to a general readership.
- 6) Materials and methods: Should be comprehensive and not simply reference a previous publication for details on how an experiment was performed. Please provide full descriptions in the text for readers who may not have access to referenced manuscripts.
- 7) All antibodies, cell lines, animals, and tools used in the manuscript should be described in full, including accession numbers for materials available in a public repository such as the Resource Identification Portal. Please be sure to provide the sequences for all of your primers/oligos and RNAi constructs in the materials and methods. You must also indicate in the methods the source, species, and catalog numbers (where appropriate) for all of your antibodies. Please also indicate the acquisition and quantification methods for immunoblotting/western blots.
- 8) Microscope image acquisition: The following information must be provided about the acquisition and processing of images:
 - a. Make and model of microscope
 - b. Type, magnification, and numerical aperture of the objective lenses
 - c. Temperature
 - d. Imaging medium

- e. Fluorochromes
- f. Camera make and model
- g. Acquisition software
- h. Any software used for image processing subsequent to data acquisition. Please include details and types of operations involved (e.g., type of deconvolution, 3D reconstitutions, surface or volume rendering, gamma adjustments, etc.).

10) Supplemental materials: There are strict limits on the allowable amount of supplemental data. * Reports may have up to 3 supplemental figures. Please also note that tables, like figures, should be provided as individual, editable files. A summary of all supplemental material should appear at the end of the Materials and methods section.

13) ORCID IDs: ORCID IDs are unique identifiers allowing researchers to create a record of their various scholarly contributions in a single place. Please note that ORCID IDs are now *required* for all authors. At resubmission of your final files, please be sure to provide your ORCID ID and those of all co-authors.

Please note that JCB now requires authors to submit Source Data used to generate figures containing gels and Western blots with all revised manuscripts. This Source Data consists of fully uncropped and unprocessed images for each gel/blot displayed in the main and supplemental figures. File names for Source Data figures should be alphanumeric without any spaces or special characters (i.e., SourceDataF#, where F# refers to the associated main figure number or SourceDataFS# for those associated with Supplementary figures). The lanes of the gels/blots should be labeled as they are in the associated figure, the place where cropping was applied should be marked (with a box), and molecular weight/size standards should be labeled wherever possible. Source Data files will be made available to reviewers during evaluation of revised manuscripts and, if your paper is eventually published in JCB, the files will be directly linked to specific figures in the published article.

Journal of Cell Biology now requires a data availability statement for all research article submissions. These statements will be published in the article directly above the Acknowledgments. The statement should address all data underlying the research presented in the manuscript. Please visit the JCB instructions for authors for guidelines and examples of statements at (<https://rupress.org/jcb/pages/editorial-policies#data-availability-statement>).

B. FINAL FILES:

**It is JCB policy that if requested, original data images must be made available to the editors. Failure to provide original images upon request will result in unavoidable delays in publication. Please ensure that you have access to all original data images prior

to final submission.**

Thank you for your attention to these final processing requirements. Please revise and format the manuscript and upload materials within 7 days. If you need an extension for whatever reason, please let us know and we can work with you to determine a suitable revision period.

Thank you for this interesting contribution, we look forward to publishing your paper in Journal of Cell Biology.

Sincerely,

Lois Weisman, PhD
Monitoring Editor

Andrea L. Marat, PhD
Deputy Editor

Journal of Cell Biology

Reviewer #1 (Comments to the Authors (Required)):

The authors have addressed my technical comments that were made in the first round of review.

Reviewer #2 (Comments to the Authors (Required)):

The authors have adequately addressed my original concerns, in part by adding clarifications to the text and also by performing additional experiments that are now included as a new Extended figure and discussed in the text. No further changes or additions are required.